# Identification and classification of factors affecting the non-use of safety harness at height among construction workers in Tehran

**Parvin Sepehr**[1☯], **Mahboobeh eshaghi**[2], **Mousa Jabbari** [1,3☯] *, **Hassan Sadeghi Naeini**[4], **Mansour Ziaei**[5], **Ali Salehi sahl abadi**[1☯]

1 Department of Occupational Health and Safety Engineering, School of Public Health and Safety, Shahid Beheshti University of Medical Sciences, Tehran, Iran, 2 Department of Occupational Health and Safety Engineering, School of Public Health and Safety, Kerman University of Medical Sciences, Kerman, Iran, 3 Workplace Health Promotion Research Center, Shahid Beheshti University of Medical Sciences, Tehran, Iran, 4 Industrial Design Department, School of Architecture & Environmental Design, Iran University of Science & Technology, Tehran, Iran, 5 Department of Health, Safety and Environment, School of Health, Bushehr University of Medical Sciences, Bushehr, Iran

☯ These authors contributed equally to this work.

* jabbarim@sbmu.ac.ir

## Abstract

### Introduction

The accident of falling from a height is high among construction workers. Construction workers do not use harnesses. Thus, the present study was conducted to identify the factors affecting the non-use of harnesses among construction workers in Tehran, Iran.

### Materials and methods

In this study was conducted by interviewing professors and construction workers in order to identify factors affecting the non-use of harness. Factors influencing the non-use of safety harnesses were identified from the workers' point of view. The obtained data were classified and coded using MAXQDA 10 software. After that, the most essential, effective and powerful factors were identified using the degree and intersectionality of social network analysis.

### Results

According to the interview results, 27 factors were determined as factors affecting the non-use of harnesses by construction workers and divided into four main groups. The four groups were harness design, management factors, harness comfort, and attitudinal factors. Based on the results of the degree centrality, the non-ergonomic design and attitude of the harness inefficiency were identified as the most influential and powerful factors. The betweenness indicator also showed that the non-ergonomic design could mediate other factors in the non-use of the harness.

**Data Availability Statement:** All relevant data are within the paper and its supporting information files.

**Funding:** Mousa Jabbari was supported by Shahid Beheshti University of Medical Sciences [31938]. The funders had no role in the study design, data collection and analysis, decision to publish, or manuscript preparation.

**Competing interests:** The authors have declared that no competing interests exist.

## Conclusion

The findings showed that by considering various factors such as considering more comfort in the design of the ergonomic harness, it produced a better product. Also, the use of safety harnesses by workers increases.

## Introduction

Each year, hundreds of construction accidents occur, resulting in both human and financial losses [1].The issue of occupational accidents in the construction sector has been a major issue for governments. Falling from height is the primary cause of fatal construction accidents in the US, Taiwan, Spain, China, Korea, and Australia, putting workers' safety at risk [2, 3]. As reported by the International Labor Organization [4], in the United States, there were 5,250 fatalities resulting from job-related causes. Of those, 731 were attributed to the construction industry, making it the second most dangerous with 33.5%, only behind motor vehicle deaths [5]. OSHA stated that the construction industry is at the forefront in terms of defects in standards[6]. Furthermore, falling from height in the construction industry has been the most important cause of death worldwide from 2001 to 2019 [7]. Jabari (2016) also showed that 57% of the causes of death and disability in Tehran construction projects were due to falls from height.[8] According to the statistics announced by the Social Security Organization of Iran in 2019, the number of 44,491 work-related accidents were recorded, of which 730 resulted in death [9]. In addition, Occupational Safety and Health Administration [10] (2018) revealed that one-fifth of accidents in the private sector are related to the construction industry Accidents caused by falling from heights impose great costs on individuals and society [11]. Thus, for preventing falling accidents related to work at height, it was recommended to use personal protective equipment (PPE) as one of the appropriate ways. In this regard, the harness has been proposed as a legal requirement for working at height. Indeed, harnesses are recommended as a last solution for eliminating the risk of falling from a height to save people's life [12]. A full-body harness is a type of body support that evenly distributes weight over the user's shoulders, hips, and thighs. To assist the worker in avoiding suspension and falls, the harnesses are designed with a D-ring.[13]Gabriel et al. [2] demonstrated the potential for incorrect harness usage by employees owing to a lack of comfort and pleasure, despite the harness being provided by the company and despite the harness being advised to be used. In addition, the study by Fang et al. demonstrated that discomfort is one of the leading causes of non-use of harnesses among construction workers who work at height.[14] Beverly et al. also showed that using a discomfort harness could lead to anxiety for the workers [15]. In another study, Kim et al. (2020) stated that scaffolders mentioned discomfort and pain as a result of using safety harnesses during work at the height [16]. Chae et al. [17] utilized a researcher-made questionnaire to examine the overall satisfaction with the use of the harness with a 7-point Likert scale. The research criteria included wearability (comfort in wearing the harness), pressure on the body, and feeling of heat and humidity; their study showed that the feeling of comfort is the main factor for its use by workers. Moreover, Angles [18] demonstrated that the main reason for the non-use of harnesses is the pressure on the workers' thighs and shoulders.

Another study showed that the majority of scaffolders were reluctant to utilize the harness. Factors such as work pressure from managers, underestimation of risk, and lack of training were identified as the main reasons [19]. Bunney et al. also demonstrated that an unergonomic

design reduces the motivation of workers to use the harness at the height [20]. Wong et al. conducted a research in which they explored the essential variables for the use or non-use of personal protective equipment among construction workers; they classified the factors for the non-use of personal protective equipment into three categories: personal, technological, and environmental. Individual face-to-face interviews with construction employees were employed to acquire data for their qualitative study [21]. In another study, Goh and Sa'adon stated that construction workers do not use harnesses due to the lack of supervisors' attitude and supervision [22]. Hsiao stated that the reason for non-use of the harness is the lack of physical fit with the harness and the user's lack of comfort with the safety harness [23].

According to the previous explanations, the scientific grounds for harness non-use are an important issue that has to be explored and assessed in numerous sectors, including construction. In this regard, qualitative studies could be used in interviews and software. Therefore, it is necessary to identify the critical factors and take corrective and preventive measures [24, 25].

In this issue, social network analysis [26] could be used. In order to find and quantify the most crucial elements based on centrality indicators in a network, the SNA is used to quantify the relationships between nodes and uncover the hidden strength of links in a network. In this research, semi-structured interviews and social network analysis methods were used to identify and categorize the reasons influencing the non-use of harnesses. Based on searches on scientific sites, very few studies have been conducted regarding the non-use of harnesses. The concept for this study was inspired by the SNA philosophy, which focuses on the interactions between each pair of actors in a network to determine how significant each player in a network is [27]. Thus, the purpose and innovation of this research focused on revealing the main causes of influencing the non-use of the harness through the SNA analysis as a quantitative approach.

## Materials and methods

This cross-sectional and qualitative study was undertaken to identify factors affecting the non-use of harnesses among construction workers through semi-structured interviews in Tehran, 2022. The study protocol was reviewed and approved by the Research Ethics Committees of the School of Public Health & Neuroscience Research Center—Shahid Beheshti University of Medical Sciences, Tehran. Iran (Approval ID: R. SBMU.PHNS.REC.1401.083). The study was conducted in a quiet place at the construction sites. Each worker was interviewed individually. The confidentiality of employees' information and opinions was upheld. Workers signed a written consent form. The aim of the research and its steps were explained to the workers. Construction workers were selected using convenience sampling focusing on available samples in the north, west, east, and south of Tehran. The interview data were classified using MAXQDA software to determine the factors affecting the non-use of harnesses. Through data analysis in this research, the data were categorized into main codes and sub-codes, respectively. After that, the critical factors were identified through the SNA.

### Identifying factors affecting the non-use of the harness

In this study, a semi-structured interview was used to identify the factors affecting the non-use of harnesses. The average, minimum, and maximum interview times were 30, 20, and 45 minutes, respectively. The mentioned times were chosen to manage the unreasonable answers of workers due to fatigue and boredom during the interview. A minimum of one year of experience working at heights on construction projects was a required criterion for entering research. A history of surgery in the abdomen, hip, or shoulder was an exclusion criteria. People participating in this study should have at least a year of experience using a harness, which means they should be able to tie one properly and have worn one while working in a real

workplace; this may greatly aid in communicating sentiments and experiences. Participants should have a healthy BMI since obesity might negatively influence how individuals feel when wearing a harness. The age range of participants was set from 20 to 50 because that is the working age range. With the consent of the workers, the interviews were recorded so that it is possible to review and re-analyze them. If participants choose not to continue, they may leave the study at any time.

During the interview, workers were asked to express their problems while using the harnesses, explain the reasons for their unwillingness to use them and express their feelings about the harnesses.

The following questions were posed to the participants:

What factors and suggestions do they have for the comfort of harnesses, and what suggestions and solutions do they have to improve the design of harnesses?

What problems do you face when working with the harness?

Does the size of the harness fit your body size?

Do you think the harnesses are well-designed? What is your opinion about the better design of harnesses?

Do you feel comfortable in the harness you are using?

How do you think the harness should be designed to make you feel more comfortable?

Do you always use a harness? If the answer is no, explain the reason for non-use of the harness?

In what cases do you prefer not to use a harness?

What suggestions and solutions do you have for improving the design of this harness?

New questions were raised in response to certain interviewees' responses. Since this study was qualitative, the discussion continued until the required information was obtained. The interview was continued until reaching data saturation, which means the lack of obtaining new information about the reasons for the non-use of the harness after the interview and the lack of obtaining a new code. In this study, 23 participants were questioned; beyond 13 participants, no more data were collected. However, 10 more persons were also questioned in order to guarantee the interview's outcomes. Also, in order to ensure the accuracy of the results, interviews with university professors were conducted in this regard. In this research, 23 workers were interviewed for ten days. After finishing the interviewees, the data was recorded, and the MAXQDA software was used to categorize and code the data.

## Data classification using MAXQDA software

The data obtained from the interview were entered into the MAXQDA software to qualitatively classify the factors affecting the non-use of the harness. In the MAXQDA, the data is organized and managed qualitatively [28]. This software divides the textual information through systematic classification, coding, classification, and sub-classification processes [29]. MAXQDA is a content analysis software used to organize and manage qualitative data. It helps in different stages of work including data collection, advanced data organization, help in data analysis, and displaying information and results in different ways. In this software, textual data are coded and classified through systematic processes. The use of software for coding qualitative data provides the possibility to simultaneously code codes, sub-codes and parts. Interviews and documents should be available in one space and be easily moved [29].

In the study of Lotfian and Maqri, health and safety policies for dealing with Covid-19 are categorized by the MAXQDA [30]. In addition, in the study of Moradi et al., MAXQDA software was used to identify the causes of occupational neck pain in teachers [31]. Moreover, Turedi and Caylan categorized the safety, security, and environmental issues by participants'

experiences in national marine policies using the MAXQDA software [32]. Furthermore, Alkhaleefah et al. used this software to promote transportation safety [33]. In addition, Salim et al. utilized the MAXQDA for managing fire in public healthcare buildings [34].

Data obtained from interviews were entered into MAXQDA, and this software simultaneously recorded texts. As a result, for each interview, a document was prepared that included demographic information on the workers as well as their texts. Every sentence of the interview texts was carefully read and then converted into different semantic units and finally to the smallest meaningful unit (code). According to each sentence, one or more new codes are created. The extracted codes are placed in the main and sub-classes based on similarity. In some cases, the code of the sentences is modified. The defined codes were also activated for the next interview; some of them were classified into one or more sub-codes. The present researchers examined this classification, and then the collected data were interpreted to determine the factors affecting the non-use of harnesses.

## Determining the factors influencing non-use of the harness using social network analysis

**Social network analysis.** Researchers have used the SNA technique to study networks [35]. The SNA has been used in various fields, such as actions in medical centers [36], natural resource management [37], crisis management [38], cooperation between emergency teams [27], etc. The SNA determines the relationship among nodes in a specific network [39] for identifying the most critical nodes [40]. A relationship between two entities in a two-mode network represents a node and an event to identify the most important and influential nodes and events [41]. To obtain the objective of the present study, the centrality indicators of the SNA, which include degree and betweenness centralities, were used.

Therefore, this research focused on understanding the main factor/factors influencing the non-use of harnesses among construction workers through the two-mode network as a quantitative technique. Semi-structured individual interviews were done in this stage to determine why the workers were hesitant to utilize the harness. The data of the SNA consisted of two distinct sets of entities, which the construction workers are as actors and influencing causes on non-use of the harness as events [42]. According to the affiliation matrix (Table 1), the workers-factors interaction network, and each row of the matrix shows a worker's affiliation with the influencing causes of non-use of the harness. This research used binary data (absent, i.e., 0.0, and present, i.e., 1.0), where "1" indicates that one cause is considered for the non-use of the harness by workers, and 0 (zero) indicates that there is no choice by workers as an affected cause.

*Degree centrality.* One of the essential centrality indicators is a degree that refers to the number of direct ties a node has with other nodes in a particular network [43]. In a two-mode

**Table 1. The two-mode network matrix for the workers and influencing causes on non-use of harness.**

| Actors | Events | | | | | |
|---|---|---|---|---|---|---|
| | $Cause_1$ | $Cause_2$ | $Cause_3$ | $Cause_4$ | . . .. | $Cause_n$ |
| $Workers_1$ | 1 | 0 | 1 | 1 | . . . | 1 |
| $Workers_2$ | 0 | 1 | 0 | 1 | . . . | 1 |
| $Workers_3$ | 1 | 1 | 0 | 1 | . . . | 0 |
| $Workers_4$ | 0 | 1 | 0 | 1 | . . . | 0 |
| . . .. | . . . | . . .. | . . .. | . . .. | . . . | . . .. |
| $Workers_n$ | 1 | 0 | 0 | 1 | . . . | 0 |

network, the degree of a specific node is the all number of times that a node is selected by events [27]; this implies that a node with more selections by events is more influential and powerful among nodes [41, 44] for controlling and leading a network [37] and reaching the objective of an organization [40]. The degree index helps to determine the most important factor/factors in the non-use of harnesses among workers.

*Betweenness centralit*. The betweenness indicator implies the total number of shortest paths between nodes that pass through a node as a bridge [43]. By its intermediation role, this indicator can manage the information flow between nodes in a particular network. Thus, all pairs of nodes in a two-mode network are connected by a node with higher betweenness centrality and vice versa [43, 45]. A factor with a powerful position presents more opportunities for obtaining the pleasure consequences due to its connections to different nodes [46]. It can act as a more powerful node than other positions in controlling the network as a mediator [35]. This index helps to identify the most critical factor/factors with a central position as a mediator in the non-use of harnesses among workers.

## Data analyses

In a two-mode social network, "worker"-"factors related to the non-use of harness" interaction network, each cell of the matrix presents a cause of non-use the harness by workers. The research data were analyzed using the UCINET program (Version 6.0). The values of centrality indicators were normalized to compare factors [47]. The values of centrality indicators are from 0 to 1 as a quantitative technique. A higher value represents the importance of a factor within a network and vic-versa.

## Results

### Data classification of the reasons for non-use of the harness using MAXQDA software

The demographic information of the participants in this study is as follows.

The average age of people was 30.6 ±5.2 and work experience in the construction industry was 8.37 ± 4.47. The average body mass index was 24.74±2.42. All the workers had an education level below diploma. 15% were illiterate, 45% could read and write, 20% had a bachelor's degree, and 10% had a diploma.

Fig 1 shows the classification of effective factors in non-use the harness by MAXQDA. The four main factors were including design factors, comfort, management, and attitude factors, each of which had some sub-codes as follows:

- The management factor as one of the main codes had four sub-codes, including lack of easy access, lack of supervision, lack of time to wear, and lack of Harness.

- The attitudinal factor as another main code had sub-codes, including time consuming to wear, restrictions on working, inconvenience in doing work, and ineffectiveness.

- Moreover, the factor of the harness comfort as the main code includes nine sub-codes, e.g., difficulty working with harness, lack of fitness with body dimensions (large), limiting the work, lack of comfort, lack of fitness with body dimensions (small), hard-wearing of harness, the feeling of pain in wearing, pressure on the thighs and testicles, pressure to the shoulders and waist.

- Finally, the design factors are determined by ten sub-codes, which include single point, aesthetics failure, non-use of the anti-pressure pad, low quality of materials, non-use of of

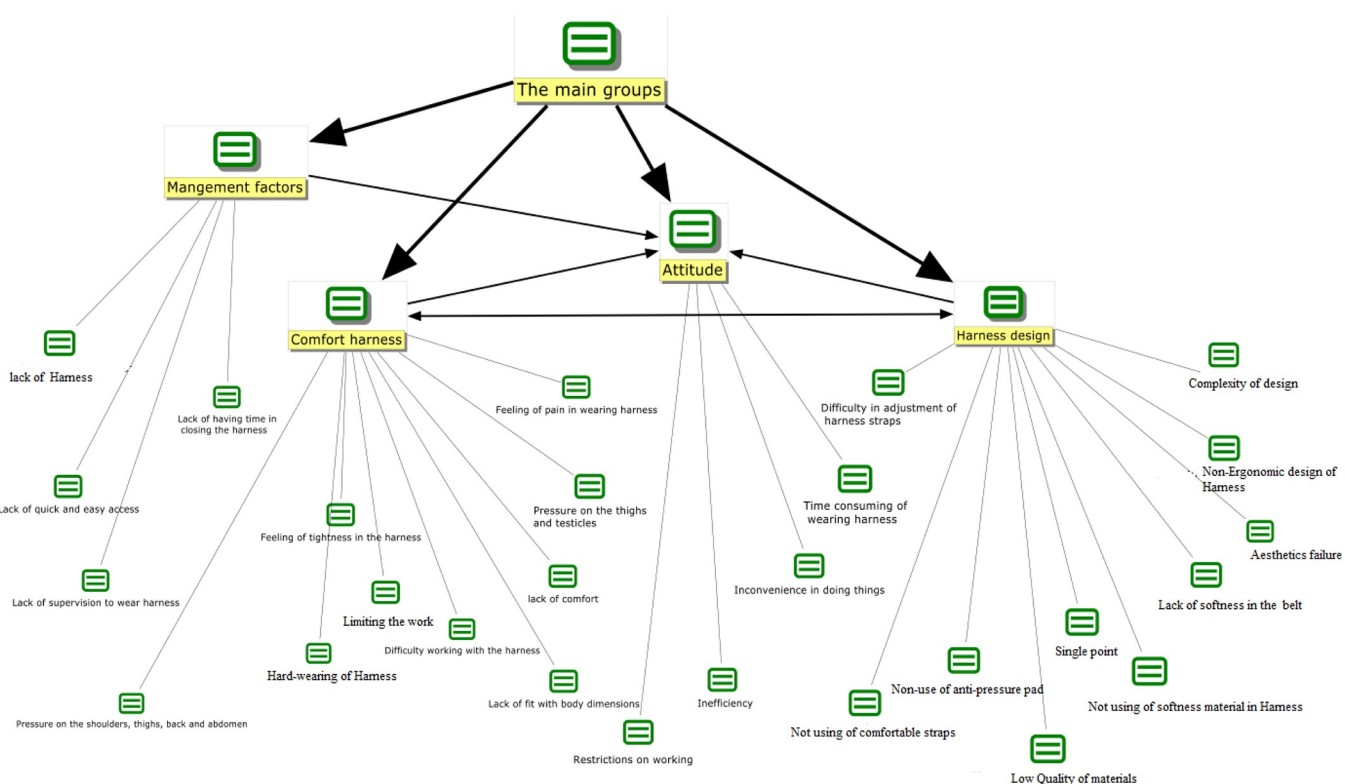

**Fig 1. Classification of effective factors in non-use the harness by MAXQDA.**

comfortable straps, complexity of design, non-use of soft material in harness, lack of softness in the belt, difficulty in the adjustment of harness straps, non-ergonomic design of harness.

## Determining the effective factors in non-use the harness b degree and betweenness centralities

According to the result (Table 2), non-ergonomic design with a degree centrality equal to 0.88 had the highest effect on workers' unwillingness to use the harnesses. This value showed three-fourths of the factors affecting non-use are directly related to the non-ergonomic design of the harness, which is related to the difference between the anthropometric variables of users and the present harness. This result is associated with the difference between the anthropometric variables of users and the current harness. In addition, 0.84 of the reasons were related to having a poor opinion of the effectiveness of the harness. Furthermore, using the betweenness centrality, it was discovered that "non-ergonomic design" (0.09), followed by "ineffectiveness" (0.06), are the two most critical reasons influencing harness non-use The result of the betweenness indicated that the non-standard design connected approximately ten percent of the other factors as an influential factor. It should be noted that the values of betweenness centrality of other factors in this study were relatively low.

Fig 2 shows the relationships between the effective factors as nodes and the selection of workers as events to understand the network better. Each symbol's size is determined based on the degree centrality value, which larger size shows the importance of that factor. In this figure, the red circle refers to the workers, and the blue square is related to factors.

**Table 2. The results of the degree and betweenness centralities of the factors in the non-use of the harness.**

|  | Factors | Centrality | |
|---|---|---|---|
| **Comfort** | | **Degree** | **Between** |
| 1 | Difficulty working with Harness | 0.24 | 0.003 |
| 2 | Lack of fitness with body Dimensions (large) | 0.24 | 0.003 |
| 3 | Limiting the work | 0.28 | 0.004 |
| 4 | Lack of comfort-others | 0.36 | 0.007 |
| 5 | Lack of fitness with body Dimensions (small) | 0.36 | 0.007 |
| 6 | Hard-wearing of Harness | 0.52 | 0.016 |
| 7 | Feeling of pain in wearing | 0.6 | 0.022 |
| 8 | Pressure on the thighs and testicles | 0.76 | 0.034 |
| 9 | Pressure to the shoulders and waist | 0.76 | 0.036 |
| **Management** | | | |
| 10 | Lack of easy access | 0.16 | 0.001 |
| 11 | Lack of supervision | 0.22 | 0.002 |
| 12 | lack of time to wear | 0.44 | 0.010 |
| 13 | lack of Harness | 0.68 | 0.027 |
| **Attitude** | | | |
| 14 | Time consuming of wearing | 0.12 | 0.001 |
| 15 | Restrictions on working | 0.44 | 0.010 |
| 16 | Inconvenience in doing work | 0.52 | 0.015 |
| 17 | Ineffectiveness | 0.84 | 0.06 |
| **Design** | | | |
| 18 | Single point [48] | 0.20 | 0.002 |
| 19 | Aesthetics failure | 0.20 | 0.003 |
| 20 | Non-use of anti-pressure pad | 0.4 | 0.010 |
| 21 | Low Quality of materials | 0.48 | 0.014 |
| 22 | Non using of comfortable straps | 0.64 | 0.025 |
| 23 | Complexity of design | 0.68 | 0.028 |
| 24 | Non using of softness material in Harness | 0.70 | 0.032 |
| 25 | Lack of softness in the belt | 0.72 | 0.032 |
| 26 | Difficulty in adjustment of harness straps | 0.76 | 0.037 |
| 27 | Non-Ergonomic design of Harness | 0.88 | 0.09 |

## Discussion

In the current study, semi-structured interviews (which typically lasted 30 minutes) were used to gather information about the variables influencing the non-use of harnesses. In this study, twenty-seven sub-groups (codes) were identified and classified into four main groups. Four main factors influence the non-use of safety harnesses, which include design, comfort, management, and attitudinal factors. In the meantime, in terms of SNA analysis, the two influencing factors in the non-use of harnesses are related to non-ergonomic design and the negative attitude of participants. The study by Hasmori et al. (2020) determined that five percent of people mentioned that the improvement of harness design could motivate users [49], which is consistent with the finding of this study. The absence of soft materials, the difficulty of wearing it, the lack of ergonomics, and the inappropriateness of the harness material are among the reasons for the design of the harness, according to the findings of the current study. In the Hasmori et al. (2020) study, the majority of the construction workers (94%) concurred that comfort was crucial when utilizing the harness [49]. In addition, the survey of this study

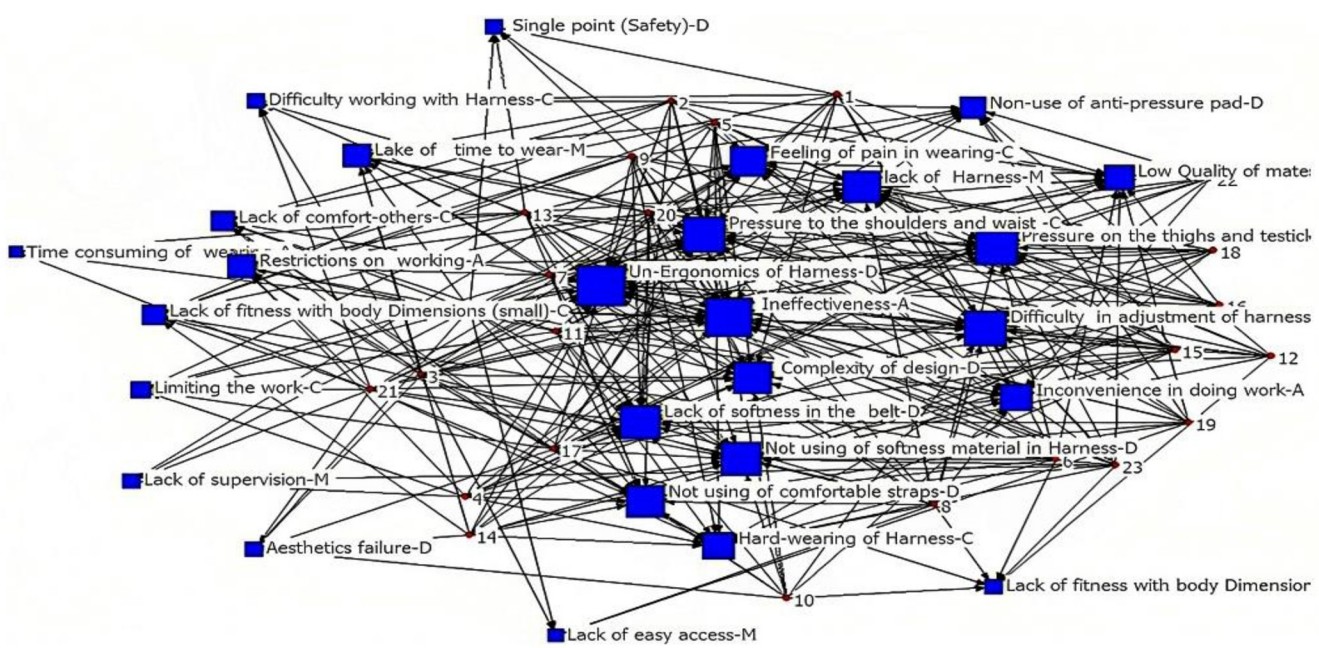

**Fig 2. The two-mode network based on the degree centrality of factors (blue square: Factors, red circle: Workers).**

showed that having a comfort sense has been important for workers during the use of harness. Moreover, Chae et al. showed that easy fastening of buckles and comfortable use of harnesses can lead to overall satisfaction among users, which was determined by semi-structured interviews. They also stated that harness design improvement and development should move toward people's comfort [17]. These findings confirmed the results of the present study. Non-ergonomic design was identified as one of the effective factors in the survey of workers. This can be attributed to employers' negligence in providing harnesses with user-friendly design. Also, the survey by Goh and Nur (2015) utilized a questionnaire through a semi-structured interview to identify the causes of non-use harnesses among scaffolders; they showed that 44% of unsafe behaviors were were mostly caused by a negative attitude on the efficacy of harnesses [22]. In the present study, attitudinal issues were raised as one of the main factors for non-use of the harness by workers. Other reasons included the harness's inconvenience, its ineffectiveness, and its restriction of work while being worn.

Additionally, Chi's research on Taiwan's construction sector revealed that 47.9% of risky behavior is associated with a low belief in the efficacy of PPEs [50]. In addition, Zhang and Fang examined 121 questionnaires regarding the reasons why Chinese scaffolders don't utilize harnesses; accepting the risk of falling from the height and having low satisfaction were idenfied as the two fundamental causes by their study [51]. In the study by Hasmori et al. (2020), 86 construction workers were asked to mention their suggestions for promoting the use of safety harnesses. The findings demonstrated discomfort during the usage, non-proving the harnesses by the employer, pressure on their body, and insufficient knowledge of how to use the harnesses as the main factors for the non-use of safety harnesses by construction workers [49]. In addition, Angles (2013) used three open questions to ask the workers' opinions about the challenges of using the harnesses and finally mentioned the poor design as one of the main factors [18]. In the present study, open questions were used to feel comfortable, express problems and develop the harness. Moreover, Hasmori et al. discovered that 53% of construction workers experience discomfort when using the harness at height [49]. According to Pisati

et al., one of the most important elements for preventing vascular thrombosis is harness comfort [52].

The findings from the degree centrality are used to identify the main factors with more powerful and influential positions for the non-use of the harness. According to the results of the degree indicator, the non-ergonomic design with the highest value was determined as the most powerful factor in the non-use of the harness in construction sites. The non-ergonomic design has a more sensitive position, as an influencing factor, than other factors. As a result, designing a standard harness according to the anthropometric dimensions of the community play an influential role in reducing unsafe behaviors and subsequently improving satisfaction at the workplace. Arteau (2018) examined the selection of full-body harnesses according to the fit and comfort of harnesses in women. Women's discomfort in working with harnesses was observed to be due to the lack of attention of harness designers to women's physical dimensions [53].

Hsiao et al. (2003) reported that traditional anthropometric data are not suitable for addressing the harness problem [23]. In addition, traditional linear anthropometric data do not correspond well to harness components and are therefore not suitable for harness design in practice [54].

Gibbons found that the carpenters who used harnesses complained about the lack of fitness and were dissatisfied with body pressure [55]. Paying attention to ergonomic issues in the design of the harness is one of the main factors in the level of comfort of people and increases the usability of the harness among users. In addition, the negative attitude to the effectiveness of the harness was determined as an influential factor. In line with this study, Prell et al. identified the powerful stakeholders in natural resource management and showed that having a higher degree is valuable for managing natural resources [37]. Moreover, Wasserman, in the study of cooperation networks, confirmed that actors with a higher degree of centrality are more active in the network and will have a more prominent role [43]. In general, non-ergonomic design and negative attitudes towards harness efficiency have influential roles in reducing the use of the harness as unsafe behavior. Therefore, paying attention to these two factors offers a valuable opportunity to improve workers' safe behaviors effectively. Based on the previous findings, the non-ergonomic design with the highest betweenness is also recognized as the most important factor bridging other factors in non-use of the harness among the workers. Studies showed that events with a higher betweenness index are more important for controlling the flow of friendship networks [43, 56] and managing natural resources among different stakeholders [37, 43]. As a result, the findings of the betweenness indicator showed that if a standard harness could be designed in compliance with the anthropometric principles, it is possible to increase users' satisfaction and help promote safe behavior.

This research was time-consuming since most of the construction workers had a low level of education and needed more explanations during the interview. In addition, due to financial and operational limitations, this research was only conducted in the capital of Iran, i.e., Tehran. Nevertheless, employers could use the results of this study to provide suitable harnesses and move toward user-centered design.

## Conclusion

This study aimed at identifying the factors that influence the non-use of harnesses. The degree and betweenness indicators of the SNA were also used in an effort to identify the elements that have the greatest impact on harness non-use. As a result, twenty-seven factors were identified and classified into four main groups. It was found that factors such as the non-ergonomic design and the ineffectiveness attitude have the highest values, implying that these factors had

powerful and influential effects on the non-use of harnesses. The findings of the SNA showed that safety behaviors could be attained by more attention to influential and powerful factors. Tehran is the capital of Iran and the most populous city of Iran. There are many constructions in this city. Also, the safety features are more in this city than in other cities. Therefore, the results of this research can reflect the conditions in Iran and the information obtained in this study can be generalized to Iran.

## Supporting information

**S1 Dataset.**
(RAR)

## Acknowledgments

The authors would like to thank all the people who cooperated in the process of conducting the research.

## Author Contributions

**Investigation:** Parvin Sepehr, Ali Salehi sahl abadi.

**Project administration:** Parvin Sepehr, Mousa Jabbari.

**Software:** Mahboobeh eshaghi.

**Supervision:** Mousa Jabbari.

**Visualization:** Hassan Sadeghi Naeini.

**Writing – review & editing:** Mansour Ziaei.

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
