## [Decision Letter · Decision Letter 0]

13 Feb 2023

PONE-D-23-00199Identification and classification of factors affecting the non-using of safety harness at height among construction workers in TehranPLOS ONE

Dear Dr. Jabbari,

Thank you for submitting your manuscript to PLOS ONE. After careful consideration, we feel that it has merit but does not fully meet PLOS ONE’s publication criteria as it currently stands. Therefore, we invite you to submit a revised version of the manuscript that addresses the points raised during the review process.

We look forward to receiving your revised manuscript.

Kind regards,

Mohammad Hossein Ebrahimi

Academic Editor

PLOS ONE

Journal Requirements:

2. In the Methods section, please provide additional details regarding participant consent. In the ethics statement in the Methods and online submission information, please ensure that you have specified what type you obtained (for instance, written or verbal, and if verbal, how it was documented and witnessed). If your study included minors, state whether you obtained consent from parents or guardians. If the need for consent was waived by the ethics committee, please include this information.

“NO - Include this sentence at the end of your statement: The funders had no role in study design, data collection and analysis, decision to publish, or preparation of the manuscript”

8. Please ensure that you refer to Figure 1 in your text as, if accepted, production will need this reference to link the reader to the figure.

Reviewers' comments:

Reviewer's Responses to Questions

**Comments to the Author**

1. Is the manuscript technically sound, and do the data support the conclusions?

Reviewer #1: Yes

Reviewer #2: Partly

Reviewer #3: Partly

Reviewer #4: Partly

Reviewer #5: Partly

Reviewer #6: Partly

2. Has the statistical analysis been performed appropriately and rigorously? 

Reviewer #1: Yes

Reviewer #2: N/A

Reviewer #3: I Don't Know

Reviewer #4: Yes

Reviewer #5: No

Reviewer #6: Yes

3. Have the authors made all data underlying the findings in their manuscript fully available?

Reviewer #1: Yes

Reviewer #2: Yes

Reviewer #3: Yes

Reviewer #4: No

Reviewer #5: Yes

Reviewer #6: Yes

4. Is the manuscript presented in an intelligible fashion and written in standard English?

Reviewer #1: Yes

Reviewer #2: No

Reviewer #3: No

Reviewer #4: No

Reviewer #5: No

Reviewer #6: Yes

5. Review Comments to the Author

Reviewer #1: The manusript is technically correct. The applied statistics have been well described. the discussion is based on the results mentioned. Recommendations by the authors are evidence based. In my opinion, I believe that the mauscript may be accepted for publication.

Reviewer #2: I would like to say that this piece of research aims to find factors affecting the non-use of safety harness at height among construction workers, has a comprehensive analysis to identify the potential and most influential factors out of all. But while reading the manuscript I found that it could be written in a much better way. I am not having the line numbers, so cannot point out the specific word or lines. Following are the comments to best of my knowledge:

1. The language used in the manuscript is not technical, only use the language used in research papers. The English grammer should be checked with a skilful person as there are quite a few mistakes in the whole document. Check each and every statement thoroughly. Try to be consistant with choice of words , like, use 'factors affecting non-use...' and not 'reasons for not using...' as you have used word 'factors' in the title and aim.

2. The introduction needs to be improved. Try not quote studies with authors names. Make your statements clear. Give rationale of your study in detail.

3. There seems no sequence in methods section. Give study design, area, population, tools, data collection, etc. in the appropriate manner.

4. Explain the data collection process in detail. Explain the selection proccedure of participants.

5. In SNA (2.3.1), it is written that the research focussed on the factors influencing the impact of non-using of harnesses. It is that way? I think the whole manuscript is about the factors affecting the non-use of harness and not influencing the impact of.. Please make it clear in the aim of the study and be consitant till the conclusion.

6. Give details of the tool used for semi-structured interview as it is not given anywhere.

7. Take second line of discussion to methods. Rewrite last four lines of first paragraph of the discussion section.

8. Make figure 2 clearer (quality wise) and simplify it, as it not understandable.

Try to work more hard on this manuscript. All the best.

Reviewer #3: Several issues need to be addressed for this research to have a significant impact.

1. I feel the Introduction could be tightened up considerably? It seems to jump around quite a bit, so I found myself having to go back over previous sections to connect them to what I was reading. It is also repetitive.

2. More description and related study in the field should be added for Social Network Analysis. It should be stated that this method of analysis is appropriate for this type of study.

3. There needs to be a much more detailed description of the sample. How was the sample selected? How large were the organizations? What percentage of the workforce is included? What is the representation of the job categories presented?

4. The lack of any discussions of the practical implications of the findings, especially in light of the potential applicability of the research, is a significant weakness.

5. References (in the text and reference list) should be prepared based on the journal format.

6. This paper would benefit from review by a native English speaker to assist with the sentence construction and spelling.

7. The manuscript is scientifically incomplete and lacks a significant, novel contribution to the field.

8. The quality of figures should be improved.

Reviewer #4: This is an interesting study. However, some major modifications are needed:

1. there are several grammatical errors throughout the passage, for example somewhere "lake" is written instead of "lack",

2. The introduction section is too short, a more extensive literature review is needed.

3. Authors wrote that a semi-quantitative approach is used. In this method there should be guide questions, which question did you use?

4. How the participants were selected? They were workers, supervisors, or others? are all of them working at height? how experienced were they? How many participants did you interview?. all these issues should be addressed in the manuscript.

5. please explain the stop point of interviews

6. section 3.1 should be upgraded by examples from your own study. examples of complex coding and categorizing make this section more elaborate.

7. there are many types of harness and PFAS in the market, did you consider the type of harness used by the participants?

Reviewer #5: Introduction

The innovation of the study needs further clarification.

Methods and results

Why were 23 people selected?

What was question in the semi-structured interview?

What characteristics did the selected people have?

What were the inclusion and exclusion criteria?

The explanations on the MAXQDA software are transferred to data analysis section.

It is better to ask the opinion of safety experts through Delphi method or interview to ensure the obtained results. Also, the literature review can be helpful for identifying the items.

The demographic characteristics of the workers should also be mentioned, especially their level of education. How is the validity of workers' opinions evaluated?

What model is used for this classification?

Some items in the design and comfort group are similar. A better classification could be done. For example, individual items, job items, organizational items, device design items, environmental items.

It is not clearly explained in method section. What was the input to the SNA model? Your study is qualitative. Where are the quantitative values entered to obtain the weights of the items in the model? Was it based on expert opinion?

More explanations are needed for the method and results section.

Discussion

In the discussion section, the obtained results need to be interpreted.

Conclusion

In the conclusion section, general results, applications of the obtained results and future studies should be mentioned.

the manuscript need the gramatical revision by native.

Reviewer #6: The present study offers useful insights to identify the factors affecting the non-using of harnesses among construction workers in Tehran, Iran. However, the manuscript should be revised in the light of following comments:

1. Methods section- The authors have not mentioned the sampling strategy adopted to select the 23 workers. The sampling design has not been explicitly mentioned. The contents/domains of the interview are also not discussed. The Study design and methods adopted are the soul of any study, The authors should carefully address this lacunae. I am sharing some articles for the reference of the authors , wherein the sampling strategy, interview domains etc are clearly mentioned. The authors may refer these for refining their manuscript.

https://link.springer.com/article/10.1186/1472-6963-14-129

https://www.frontiersin.org/articles/10.3389/fpubh.2022.870880/full

https://bmcpublichealth.biomedcentral.com/articles/10.1186/1471-2458-11-871

2. Picture quality of Figure 2 can be improved

3. Discussion section- Can the results of the study be generalized to other locations of Iran? If No, then can it be acknowledged as a limitation by the authors.

4. Typing errors- The authors should check the manuscript for typing errors and grammatical consistency.

6. PLOS authors have the option to publish the peer review history of their article (what does this mean?). If published, this will include your full peer review and any attached files.

Reviewer #1: **Yes: **Rahul Gupta

Reviewer #2: **Yes: **Reetu Paasi

Reviewer #3: No

Reviewer #4: No

Reviewer #5: No

Reviewer #6: **Yes: **TANVI KIRAN

---

## [Author Response · Author response to Decision Letter 0]

5 Apr 2023

Dear Mohammad Hossein Ebrahimi 

Editorial Section Manager  

PLOS ONE

Thank you for your attention in reviewing our paper and providing valuable comments. We have carefully reviewed your comments and have revised the manuscript accordingly. Here are some of the changes we have made in response to the reviewers' and editorial comments and questions.

Editor comments:

Comment:

Answer:

Our paper followed all PLOS ONE style guidelines, including those pertaining to file naming. We confirm that our manuscript adheres to the style requirements of the PLOS ONE, including file-naming conventions. -----------------------------------------------------------

Comment:

2. In the Methods section, please provide additional details regarding participant consent. In the ethics statement in the Methods and online submission information, please ensure that you have specified what type you obtained (for instance, written or verbal, and if verbal, how it was documented and witnessed). If your study included minors, state whether you obtained consent from parents or guardians. If the need for consent was waived by the ethics committee, please include this information. These items were added and highlighted in the method.

Answer:

The study protocol was reviewed and approved by the Research Ethics Committees of the School of Public Health and Neuroscience Research Center, Shahid Beheshti University of Medical Sciences, Tehran. Iran, with Approval ID: R. SBMU.PHNS.REC.1401.083. All the workers who participated in this study completed a written consent form. The age requirement for inclusion in the study was 20 to 50 years old. These items were added and highlighted in the method.

Comment:

“NO - Include this sentence at the end of your statement: The funders had no role in study design, data collection and analysis, decision to publish, or preparation of the manuscript”

Answer:

Mousa Jabbari was supported by Shahid Beheshti University of Medical Sciences [31938]. The funders had no role in the study design, data collection and analysis, decision to publish, or manuscript preparation

Comment:

Answer:

"All data is mentioned in the article. The tables and figures in this article contain all of the information provided. Construction industry workers were used in this study, and their information was completely confidential. Participants were not asked to provide their first or last names to ensure anonymity and reduce any potential bias in their responses. These measures were added to the methodology section and highlighted to provide clarity to readers. The inclusion of these measures was intended to address potential concerns related to participant privacy."

These items were added and highlighted in the method. These items were added to cover the latter.

Comment:

Answer:

There is no change in the data availability statement. All the data are mentioned in the article. The tables and figures in the article are the entire information of the article.

Comment:

Answer:

 ORCID iD is announced.

Comment:

Answer:

"All the workers who participated in this study completed a written consent form. The work steps were explained to the participants. The study protocol was reviewed and approved by the Research Ethics Committees of the School of Public Health & Neuroscience Research Center at Shahid Beheshti University of Medical Sciences in Tehran, Iran, with Approval ID: R. SBMU.PHNS.REC.1401.083. These items were added and highlighted in the methods section. We thank you for your attention and for reviewing our paper, and for providing valuable comments."

Reviewer #1: The manusript is technically correct. The applied statistics have been well described. the discussion is based on the results mentioned. Recommendations by the authors are evidence based. In my opinion, I believe that the mauscript may be accepted for publication

Answer:

Thank you for your opinion and kind attention, dear Reviewer. Dear Reviewer, I appreciate your insight and thoughtful consideration.

Reviewer #2: I would like to say that this piece of research aims to find factors affecting the non-use of safety harness at height among construction workers, has a comprehensive analysis to identify the potential and most influential factors out of all. But while reading the manuscript I found that it could be written in a much better way. I am not having the line numbers, so cannot point out the specific word or lines. Following are the comments to best of my knowledge:

Comment:

. 1. The language used in the manuscript is not technical, only use the language used in research papers. The English grammer should be checked with a skilful person as there are quite a few mistakes in the whole document. Check each and every statement thoroughly. Try to be consistant with choice of words , like, use 'factors affecting non-use...' and not 'reasons for not using...' as you have used word 'factors' in the title and aim.

Answer:

We thank you for your attention and reviewing our paper and presenting valuable and insightful comments. According to the opinion of the respected referee, corrections were made and highlighted in the text.

Comment:

2. .The introduction needs to be improved. Try not quote studies with authors names. Make your statements clear. Give rationale of your study in detail.

Answer:

Introduction improved. More literature review was added. These items were added and highlighted in the method .OSHA states that the construction industry is at the forefront in terms of defects in the standards. Accidents caused by falls from heights impose significant costs on both individuals and society. A full-body harness is a body support device that distributes forces between the wearer's shoulders, hips, and thighs. The harnesses have a D-ring in their design to help workers prevent falls and suspension. Moreover, a study by Angles demonstrated that the main reason for not using harnesses was the pressure on the workers' thighs and shoulders. Another study showed that most scaffolders were less willing to use the harness. Factors such as work pressure from managers, risk underestimation, and lack of training were identified as the main reasons. Bunney et al. also demonstrated that an unergonomic design reduces the motivation of workers to use harnesses at height. Wong et al. conducted a study on critical factors for the use or non-use of personal protective equipment among construction workers. They are influenced by personal, technological, and environmental factors. In their qualitative study, they conducted individual face-to-face interviews with Hong Kong construction workers to collect data. Goh and Sa'adon stated in their study that construction workers do not use harnesses due to the lack of supervisors' attitude and supervision. Hsiao stated that the reason for not using a harness is the lack of physical fit with the harness and the user's lack of comfort from the safety harness. The highlighted items have been added to the Methods section.

Comment:

3. .There seems no sequence in methods section. Give study design, area, population, tools, data collection, etc. in the appropriate manner.

Answer:

Construction workers were selected using convenience sampling, focusing on available samples in the north, west, east, and south of Tehran. These items were added and highlighted in the method. This is a qualitative study. Therefore, the sample size has been used to check the available samples and data saturation, as presented in the method. The interview was continued until the information obtained reached saturation, or in other words until no new information about the reasons for not using the harness was obtained after the interview.

.

Comment:

4. .Explain the data collection process in detail. Explain the selection proccedure of participants

Answer:

Construction workers were selected using convenience sampling, focusing on available samples in the north, west, east, and south of Tehran. 

In this study, a semi-structured interview was used to identify the factors affecting the non-using of harnesses. The average, minimum, and maximum interview times were 30, 20, and 45 minutes, respectively. This time was chosen to manage the unreasonable answers of workers due to fatigue and boredom during the interview. One of the criteria for entering workers into the study was having at least one year of experience working at height in construction projects. Then, the workers were asked to explain the factors affecting the non-use of harnesses. With the consent of the workers, the interviews were recorded so that it was possible to review and reanalyze them. In the interview with the workers, they were asked to express their problems while using the harnesses and explain the reasons for not wanting to use it and express their feelings about the harnesses, what factors and suggestions they have for the comfort of harnesses, and what suggestions and solutions they have to improve the design of harnesses. New questions were raised based on the answers of some interviewees. Because this study was qualitative, the discussion continued until the required information was obtained. In this study, 23 workers were interviewed over ten days. After completing the interviews, the data were recorded, and MAXQDA software was used to categorize and code the data. These items were added and highlighted in the method.

Comment:

. 5. In SNA (2.3.1), it is written that the research focussed on the factors influencing the impact of non-using of harnesses. It is that way? I think the whole manuscript is about the factors affecting the non-use of harness and not influencing the impact of. Please make it clear in the aim of the study and be consitant till the conclusion.

Answer:

The main aim of the study was to identify the factors affecting the non-use of harnesses among construction workers, as mentioned in the Abstract and Methods sections. According to your comment, we have removed the impact of the manuscript as follows:

Therefore, this study focused on understanding the main factors influencing the non-use of harnesses among construction workers through a two-mode network as a quantitative technique. These items were added and highlighted in the method.

Comment

. 6. Give details of the tool used for semi-structured interview as it is not given anywhere.

Answer:

Based on the studies, some questions were prepared. Questions were asked to the workers and other questions arose from the answers of the workers that were asked to the people. The 

Based on these studies, several questions have been prepared. Questions were asked to the workers and other questions arose from the answers of the workers that were asked to the people. The participants were asked the following questions:

What problems do you face when working with a harness?

Does the size of the harness fit your body size?

Do you think the harnesses are well-designed? What is your opinion about a better design of harnesses?

Do you feel comfortable with the harness you use?

How do you think a harness should be designed to make you feel more comfortable?

Do you always use harnesses? If the answer is no, explain the reason for not using the harness

In what cases do you prefer to not use harnesses?

What suggestions and solutions do you have to improve the design of this harness ?

These questions items were added and highlighted in the method

Comment

7. .Take second line of discussion to methods. Rewrite last four lines of first paragraph of the discussion section.

Answer:

The second line of discussion concerns the methods.In the present study, the factors affecting non-use the harnesses were investigated through a semi-structured interview (the average interview duration was 30 minutes). Four main factors influence the non-use of safety harnesses: design, comfort, management, and attitude. In the meantime, in terms of SNA analysis, the two influencing factors in non using restraints are related to non-ergonomic design and negative attitude of people.These items were added highlighted in the discussion

It was stated in the method.

Comment:

8. Make figure 2 clearer (quality wise) and simplify it, as it

Answer:

A: We tried to enhance the resolution of Figure 2 as much possible as. These items were added and highlighted in the result.

Reviewer #3

Comment

1. I feel the Introduction could be tightened up considerably? It seems to jump around quite a bit, so I found myself having to go back over previous sections to connect them to what I was reading. It is also repetitive.

Answer:

Introduction improved. More literature review was added. These items were added and highlighted in the method .OSHA states that the construction industry is at the forefront in terms of defects in the standards. Accidents caused by falls from heights impose significant costs on both individuals and society. A full-body harness is a body support device that distributes forces between the wearer's shoulders, hips, and thighs. The harnesses have a D-ring in their design to help workers prevent falls and suspension. Moreover, a study by Angles demonstrated that the main reason for not using harnesses was the pressure on the workers' thighs and shoulders. Another study showed that most scaffolders were less willing to use the harness. Factors such as work pressure from managers, risk underestimation, and lack of training were identified as the main reasons. Bunney et al. also demonstrated that an unergonomic design reduces the motivation of workers to use harnesses at height. Wong et al. conducted a study on critical factors for the use or non-use of personal protective equipment among construction workers. They are influenced by personal, technological, and environmental factors. In their qualitative study, they conducted individual face-to-face interviews with Hong Kong construction workers to collect data. Goh and Sa'adon stated in their study that construction workers do not use harnesses due to the lack of supervisors' attitude and supervision. Hsiao stated that the reason for not using a harness is the lack of physical fit with the harness and the user's lack of comfort from the safety harness. The highlighted items have been added to the Methods section.

Comment: 

2. More description and related study in the field should be added for Social Network Analysis. It should be stated that this method of analysis is appropriate for this type of study.

Answer:

A: According to the comment, we added the following paragraph in the main manuscript.

The idea of this research was initiated in the philosophy of the SNA that focuses on the relationships between each pair of actors in a network, which specifies how important an actor in a network is. Thus, this research focused on revealing the main causes of influencing the non-use of the harness through the SNA analysis as a quantitative approach.

In this step, to gain an understanding of why the workers were unwilling to use the harness, semi-structured individual interviews were conducted. The data of the SNA consisted of two distinct sets of entities, which the construction workers are as actors and influencing causes on non-use of the harness as events. According to the affiliation matrix (table 1), the workers-causes interaction network, and each row of the matrix shows a worker’s affiliation with the influencing causes of non-use of the harness. This research used binary data (absent, i.e. 0.0, and present, i.e. 1.0) that "1" means that one cause is considered for the non-use of the harness by workers. In addition, a 0 (zero) indicated that was no selection by workers as an affected cause.

Table 1. The two-mode network matrix for the workers and influencing causes on non-use of harness

Actors Events

 Cause1 Cause2 Cause3 Cause4 …. Causen

Workers1 1 0 1 1 … 1

Workers2 0 1 0 1 … 1

Workers3 1 1 0 1 … 0

Workers4 0 1 0 1 … 0

…. … …. …. …. … ….

Workersn 1 0 0 1 … 0

Comment 3. There needs to be a much more detailed description of the sample. How was the sample selected? How large were the organizations? What percentage of the workforce is included? What is the representation of the job categories presented?

Answer:

This is a qualitative study. Therefore, the sample size has been used to check the available samples and data saturation, as presented in the method. The interview was continued until the information obtained reached saturation, or in other words until no new information about the reasons for not using the harness was obtained after the interview.

Comment

4. The lack of any discussions of the practical implications of the findings, especially in light of the potential applicability of the research, is a significant weakness.

Answer: 

The Discussion section has been revised according to the opinion of the reviewer.

These items have been added and highlighted in the discussion.

In the present study, the factors affecting non-use of the harnesses were investigated through a semi-structured interview (the average interview duration was 30 minutes). Four main factors influence the non-use of safety harnesses, which include design, comfort, management, and attitudinal factors. In the meantime, in terms of SNA analysis, the two influencing factors in non-use of restraints harnesses are related to non-ergonomic design and negative attitude of people. According to the findings of the present study, one of the reasons related to the design of the harness is the lack of use of soft materials in the harness, the complexity of wearing it, the lack of ergonomics, and the inappropriateness of the harness material. Non-ergonomic design was identified as one of the effective factors in the survey of workers. This can be attributed to employers' negligence in providing harnesses with user-friendly design. In the present study, attitudinal issues were raised as one of the main factors and groups in people's non-use of the harness, and factors such as the cumbersomeness of the harness, the limitation of work by wearing the harness, and its ineffectiveness were pointed out. In the present study, open questions were used in order to feel comfortable, express problems and develop the harness

Comment

5. References (in the text and reference list) should be prepared based on the journal format.

Answer:

References was prepared based on the journal format.

Comment

6. This paper would benefit from review by a native English speaker to assist with the sentence construction and spelling.

Answer:

This study uses a review by a native English speaker to assist with sentence construction and spelling.

7. The manuscript is scientifically incomplete and lacks a significant, novel contribution to the field.

Based on searches on scientific sites, very few studies have been conducted regarding the non-use of harnesses. The purpose and innovation of this study is to reveal the main causes influencing the non-use of restraint through the analysis of SNA as a focused quantitative approach. These items were added and highlighted in the Introduction.

Reviewer #4: 

This is an interesting study. However, some major modifications are needed:

Comment:

1. there are several grammatical errors throughout the passage, for example somewhere "lake" is written instead of "lack",

Answer:

We thank you for your attention and reviewing our paper and presenting valuable comments.

Items were corrected.

Comment:

2. The introduction section is too short, a more extensive literature review is needed.

Answer:

Introduction improved. More literature review was added. These items were added and highlighted in the method. OSHA states that the construction industry is at the forefront in terms of defects in standards. Accidents caused by fall from height impose great costs on individuals and society. A full-body harness is a body support device that distributes forces between the wearer's shoulders, hips and thighs. The harnesses have a D-ring in their design to help the worker prevent fall and suspension .Moreover, the study by Angles demonstrated that the main reason for non using harnesses was the pressure on the workers' thighs and shoulders. Another study showed that most scaffolders were less willing to use the harness. Factors such as work pressure from managers, underestimation of risk, and lack of training were identified as the main reasons. Bunney et al. also demonstrated that an unergonomic design reduces the motivation of workers to use the harness at the height.Wong et al. in a study on critical factors for the use or non-use of personal protective equipment among construction worker .It is influenced by personal, technological and environmental factors. In their qualitative study, they used individual face-to-face interviews with Hong Kong construction workers to collect data. In their study, Goh and Sa'adon stated that construction workers do not use harnesses due to the lack of supervisors' attitude and supervision. Hsiao states the reason for non using the harness is the lack of physical fit with the harness and the user's lack of comfort from the safety harness. 

Comment:

3. Authors wrote that a semi-quantitative approach is used. In this method there should be guide questions, which question did you use?

Answer:

In the interview with the workers, they were asked to express their problems while using the harnesses and explain the reasons for not wanting to use it and express their feelings about the harnesses.What factors and suggestions do they have for the comfort of harnesses and what suggestions and solutions do they have to improve the design of harnesses. The following questions were asked to the people.

(1) What problems do you face when working with the harness?

(2) Does the size of the harness fit your body size?

(3) Do you think the harnesses are well designed? What is your opinion about the better design of harnesses?

(4) Do you feel comfortable in the harness you are using?

(5) How do you think the harness should be designed to make you feel more comfortable?

(6) Do you always use a harness? If the answer is no, explain the reason for not using the harness

(7) In what cases do you prefer not to use a harness?

What suggestions and solutions do you have for improving the design of this harness.

These questions items were added and highlighted in the method

Comment:

4. How the participants were selected? They were workers, supervisors, or others? are all of them working at height? how experienced were they? How many participants did you interview?. all these issues should be addressed in the manuscript.

Answer:

Participants aged 20–50 years were included in this study. All the workers who participated in this study completed written consent forms. The work steps were explained to the workers. Construction workers were selected using convenient sampling focusing on available samples in the north, west, east and south of Tehran.

One of the criteria for entering workers into the study was having at least one year of experience working at height in construction projects. In this study, simple construction workers with at least one year of experience working with harnesses were selected. Construction supervisors do not work at height, only simple workers work at height. These items were added and highlighted in the method.

Comment:

5. please explain the stop point of interviews

Answer:

The interview was continued until the information obtained reached saturation, or in other words until no new information about the reasons for not using the harness was obtained after the interview. These items were added and highlighted in the work method.

Comment:

6. section 3.1 should be upgraded by examples from your own study. examples of complex coding and categorizing make this section more elaborate.

Answer:

Examples from previous studies and coding are also provided. These items were added and highlighted in the method.

In addition, in the study by Mardadi et al., the MAXQDA software was used to identify the causes of occupational neck pain in teachers [20]. Moreover, Turedi and Caylan categorized safety, security, and environmental issues based on participants' experiences in national marine policies using the MAXQDA software [21]. Alkhaleefah et al. used this software to promote transportation safety[22]. In addition, Salim et al. utilized MAXQDA to manage fires in public healthcare buildings[23].

According to the opinion of the respected Reviewer, the discussion section was revised.

These items were added and highlighted in the discussion.

Comment:

7. there are many types of harness and PFAS in the market, did you consider the type of harness used by the participants?

Answer:

In many construction workshops in Iran, harness is not used. In some construction workshops, the simplest type of restraint is also used. The results of our surveys showed that most people use the simplest type of harness, which is the full body harness class A for fall arrest, designed to protect the body during and after the fall arrest, they must have at least one fall arrest connection element that They should be explained in such a way that it is placed on the back in the middle of the rib cage, in the middle of the chest, approximately at the height of the sternum of the person who wears it. Simple harnesses are harnesses that a person does not work with in a suspended manner and only protects him in case of a fall. Therefore, under normal conditions, there is no pressure on the worker. We used this type of harnesses in this research and they do not have cushions and thick straps.

Reviewer #5: 

Comment:

1. Why were 23 people selected?

Answer:

Since this part of the study was qualitative, the interview was continued until the information obtained reached saturation, or in other words, until a new code was obtained. In this study, 23 people were interviewed and no new information was obtained after 13 people. However, to ensure the results of the interviews, ten other people were also interviewed. In addition, to ensure the accuracy of the results, interviews were conducted with university professors. These items were added and highlighted in the method.

Comment:

2. What was question in the semi-structured interview?

Answer:

In the interviews with the workers, they were asked to express their problems while using the harnesses, explain the reasons for not wanting to use them, and express their feelings about the harnesses. What factors and suggestions do they have regarding the comfort of harnesses? What suggestions and solutions do they need to improve the design of harnesses?

Comment:

What characteristics did the selected people have?

Answer:

People in the age range of 20 to 50 years were included in this study. One of the criteria for entering workers into the study was having at least one year of experience working at height in construction projects.

Comment:

What were the inclusion and exclusion criteria?

Answer:

.

One of the criteria for entering workers into the study was having at least one year of experience working at height in construction projects. The exclusion criterion was a history of surgery in the abdominal, hip, and shoulder areas. These items were added and highlighted in the method.

Comment:

The demographic characteristics of the workers should also be mentioned, especially their level of education?

Answer:

We thank you for your attention and reviewing our paper and presenting valuable comments. Demographic information was added to the text. The average age of people was 30.6 ±5.2 and work experience in the construction industry was 8.37 ± 4.47. The average body mass index was 24.74±2.42. All the workers had an education level below diploma. 15% were illiterate, 45% could read and write, 20% had a bachelor's degree, 10% had a diploma, and 10% had a diploma. . These items were added and highlighted in the 

Comment:

It is not clearly explained in method section. What was the input to the SNA model? Your study is qualitative. Where are the quantitative values entered to obtain the weights of the items in the model? Was it based on expert opinion?

Answer:

. In this step, to gain an understanding of why the workers were unwilling to use the harness, semi-structured individual interviews were conducted. The data of the SNA consisted of two distinct sets of entities, which the construction workers are as actors and influencing causes on non-use of the harness as events[29]. According to the affiliation matrix (table 1), the workers-causes interaction network, and each row of the matrix shows a worker’s affiliation with the influencing causes of non-use of the harness. This research used binary data (absent, i.e. 0.0, and present, i.e. 1.0) that "1" means that one cause is considered for the non-use of the harness by workers. In addition, a 0 (zero) indicated that was no selection by workers as an affected cause. These items were added and highlighted in the method.

Reviewer #6:

Comment:

The present study offers useful insights to identify the factors affecting the non-using of harnesses among construction workers in Tehran, Iran

Answer:

We thank you for your attention.

Comment:

1. Methods section- The authors have not mentioned the sampling strategy adopted to select the 23 workers. The sampling design has not been explicitly mentioned.

Answer:

.Since this part of the study was qualitative, the interview was continued until the information obtained reached saturation, or in other words, until a new code was obtained. In this study, 23 people were interviewed, and no new information was obtained after 13 people. But in order to ensure the results of the interview, 10 other people were also interviewed. Also, in order to ensure the accuracy of the results, interviews with university professors were also conducted in this regard. Construction workers were selected using convenient sampling focusing on available samples in the north, west, east and south of Tehran.

Comment:

3. Discussion section- Can the results of the study be generalized to other locations of Iran?

Answer:

The information obtained from this study can be generalized to Iran. Tehran is the capital and the most populous city in Iran. There are many construction projects in the city. In addition, the safety features in this city were greater than those in other cities. Therefore, the results of this research reflect the conditions in Iran. Due to a large number of construction activities in Tehran, people from other cities also refer to Tehran for employment; as a result, the results can be generalized to the whole of Iran. These items were added and highlighted in the Conclusion.

Thank you for your valuable comments.

Sincerely Yours

Dr. Mousa Jabbari

---

## [Decision Letter · Decision Letter 1]

18 May 2023

PONE-D-23-00199R1Identification and classification of factors affecting the non-use of safety harness at height among construction workers in TehranPLOS ONE

Dear Dr. Jabbari,

Thank you for submitting your manuscript to PLOS ONE. After careful consideration, we feel that it has merit but does not fully meet PLOS ONE’s publication criteria as it currently stands. Therefore, we invite you to submit a revised version of the manuscript that addresses the points raised during the review process.

We look forward to receiving your revised manuscript.

Kind regards,

Mohammad Hossein Ebrahimi

Academic Editor

PLOS ONE

Reviewers' comments:

Reviewer's Responses to Questions

**Comments to the Author**

1. If the authors have adequately addressed your comments raised in a previous round of review and you feel that this manuscript is now acceptable for publication, you may indicate that here to bypass the “Comments to the Author” section, enter your conflict of interest statement in the “Confidential to Editor” section, and submit your "Accept" recommendation.

Reviewer #2: (No Response)

Reviewer #3: All comments have been addressed

Reviewer #4: All comments have been addressed

Reviewer #7: (No Response)

2. Is the manuscript technically sound, and do the data support the conclusions?

Reviewer #2: Partly

Reviewer #3: Partly

Reviewer #4: Yes

Reviewer #7: Yes

3. Has the statistical analysis been performed appropriately and rigorously? 

Reviewer #2: I Don't Know

Reviewer #3: Yes

Reviewer #4: Yes

Reviewer #7: Yes

4. Have the authors made all data underlying the findings in their manuscript fully available?

Reviewer #2: Yes

Reviewer #3: Yes

Reviewer #4: Yes

Reviewer #7: Yes

5. Is the manuscript presented in an intelligible fashion and written in standard English?

Reviewer #2: No

Reviewer #3: Yes

Reviewer #4: Yes

Reviewer #7: (No Response)

6. Review Comments to the Author

Reviewer #2: Although many comments have been addressed but it has been repeatedly asked to carefully check the whole manuscript for grammer and use only appropriate research language. The document still lacks that scientific writing. The staements are not connecting at many places and there are repetitions; whether it is introduction or conclusion.

There are many minor mistakes which should not be present at this point of time, like, the first paragarph of results has a repetition, please correct them.

Reviewer #3: - Cohesion and coherence of sentences (paragraphs) should be considered in the introduction section.

- Relevant references should be added to sentences. For instance, in the introduction section, according to the previous explanations, according to the studies, and ... need references.

Reviewer #4: The manuscript is acceptable based on the performed modifications. Howover, I can not approve that the manuscript is written and organized based on the journal guidlines.

Reviewer #7: Dear author,

Thank you for the efforts of the respected author in writing the manuscript. A few points are raised, please note and correct:

• Is it better to mention the statistics of occupational accidents in Iran, especially falling from a height, in the introduction? And that it should be explained how the situation of such incidents is in Iran.

• It should be explained whether other articles, regulations and references have been checked in extracting the factors, or whether it has been considered only by experts? This is very important.

• The benefits of using the MAXQDA software in this study will be explained more. Of course, this is one of the strengths of the study, but it is better to explain it more.

• If possible, the quality of the Fig 2 should be improved.

• More articles need to be compared in the discussion.

• In which of the factors is the position of the cost of buying harness considered?

• In which factors is the place of education and instructions important?

• Inclusion & exclusion criteria are not clearly stated. It is necessary to complete this part by mentioning the reasons.

• Ethical considerations need to be mentioned in detail.

• The abstract needs to be organized and completed.

• The text needs to be edited by a language expert for ease of reading.

7. PLOS authors have the option to publish the peer review history of their article (what does this mean?). If published, this will include your full peer review and any attached files.

Reviewer #2: **Yes: **Dr Reetu Passi

Reviewer #3: No

Reviewer #4: No

Reviewer #7: No

---

## [Author Response · Author response to Decision Letter 1]

8 Jun 2023

Dear Mohammad Hossein Ebrahimi 

Editorial Section Manager  

PLOS ONE

We thank you for your attention and reviewing our paper and presenting valuable comments.

We read the comments precisely and revised the manuscript. Some of the changes and answers to reviewers and editorial questions and comments are as follows: 

Reviewer #2 

Comment:

1. Although many comments have been addressed but it has been repeatedly asked to carefully check the whole manuscript for grammer and use only appropriate research language. The document still lacks that scientific writing. The staements are not connecting at many places and there are repetitions; whether it is introduction or conclusion.

There are many minor mistakes which should not be present at this point of time, like, the first paragarph of results has a repetition, please correct them.

Answer:

Thanks to the reviewer, the whole manuscript was re-checked for grammar.

Duplicate sentences like the following sentence were removed.

In this study, four main groups and twenty-seven sub-codes are determined regarding the unwillingness of workers to use of harnesses.

The average body mass index was 24.74±2.42. All the workers had an education level below diploma. 15% were illiterate, 45% could read and write, 20% had a bachelor's degree, and 10% had a diploma.

According to the opinion of the respected Reviewer, changes were made in the introduction and discussion for the purpose of greater coherence.

Reviewer #3: 

Comment:

- Cohesion and coherence of sentences (paragraphs) should be considered in the introduction section.

- Relevant references should be added to sentences. For instance, in the introduction section, according to the previous explanations, according to the studies, and ... need references.

Answer:

According to the opinion of the respected Reviewer, references were added to sentences without references.

In this regard, qualitative studies could be used in interviews and software. Therefore, it is necessary to identify the critical factors and take corrective and preventive measures[24, 25].

These items were added and highlighted .

Reviewer #4:

Comment:

The manuscript is acceptable based on the performed modifications. Howover, I can not approve that the manuscript is written and organized based on the journal guidlines.

Answer:

The manuscript was completed according to the format of the magazine and tried to apply all the desired items.

Reviewer #7

Comment:

• Is it better to mention the statistics of occupational accidents in Iran, especially falling from a height, in the introduction? And that it should be explained how the situation of such incidents is in Iran.

Answer:

Thanks for the comments of the respected Reviewer, Jabari (2016) also showed that 57% of the causes of death and disability in Tehran construction projects were due to falls from height. According to the statistics announced by the Social Security Organization of Iran in 2019, the number of 44,491 work-related accidents was recorded, of which 730 resulted in death. These items were added and highlighted in the Introduction.

Comment:

• It should be explained whether other articles, regulations and references have been checked in extracting the factors, or whether it has been considered only by experts? This is very important.

Answer:

The obtained factors were based on the opinions of workers, experts and professors.

According to the answers of some interviewees, new questions were raised. Since this study was qualitative, the discussion continued until the required information was obtained. The interview was continued until reaching data saturation, which means the lack of obtaining new information about the reasons for the non-use of the harness after the interview and the lack of obtaining a new code. In this study, 23 people were interviewed, and no new information was obtained after 13 people. But in order to ensure the results of the interview, 10 other people were also interviewed. Also, in order to ensure the accuracy of the results, interviews with university professors were conducted in this regard. In this research, 23 workers were interviewed for ten days. These items highlighted in the method.

Comment:

• More articles need to be compared in the discussion

Answer:

Thanks for the comments of the respected Reviewer.

In the ARTEAU study in 2018, he examined the selection of full-body harnesses according to the fit and comfort of harnesses in women. Women's discomfort in working with harnesses is due to the lack of attention of harness designers to women's physical dimensions.

Hsiao et al. (2003) reported that traditional anthropometric data are not suitable in addressing the harness problem. 

In addition, traditional linear anthropometric data do not correspond well to harness components and are therefore not suitable for harness design in practice.

Gibbons found that the carpenters who used harnesses complained about the lack of fitness and were dissatisfied with body pressure Paying attention to ergonomic issues in the design of the harness is one of the main factors in the level of comfort of people and increases the usability of the harness among users.

These items were added and highlighted in the discussion.

Comment:

If possible, the quality of the Fig 2 should be improved.

Answer:

We tried to enhance the resolution of Figure 2 as much possible as. These items were added 

Comment:

• In which of the factors is the position of the cost of buying harness considered?

• In which factors is the place of education and instructions important?

Answer:

The opinion of the respected Reviewer regarding the cost of the harness and training to use the harness is very valuable.

This article is a part of the harness design project, and the purpose of this study was to examine the lack of use of harness by workers from the perspective of experts. As a rule, according to Iran's labor law, the employer is obliged to provide personal protective equipment and teach how to use it correctly. And in this study, no person mentioned the lack of training regarding the use of harness.

Certainly, according to other studies such as Muhammad Fikri Hasmori et al.'s study entitled Causes for Lack of Usage of Safety Harness among Construction Workers in Malaysia: An Investigation, it has addressed more reasons.

Comment:

Inclusion & exclusion criteria are not clearly stated.

Answer:

The criteria for entering workers into the study was having at least one year of experience working at height in construction projects. The BMI of the people should be in the normal range. The age of the participants should be between 20 and 50 years. The exclusion criterion was having a history of any surgery in the abdomen, hip, and shoulder area. If the participants do not want to, they can leave the study during the study. In this study, people should have at least one year of experience using a harness, that is to know at least how to properly tie a harness and also have experience while working and wearing a harness in a real work environment. Because this can help a lot in conveying feelings and experiences. People should have a suitable BMI because people's obesity may affect people's bad feeling by the harness. And the age range of people was chosen between 20 and 50 years old, which is the age range for work. These criteria have also been used in studies.

These items were added and highlighted in the method.

Comment:

• The benefits of using the MAXQDA software in this study will be explained more. Of course, this is one of the strengths of the study, but it is better to explain it more.

Answer:

MAXQDA is a content analysis software used to organize and manage qualitative data. It helps in different stages of work including data collection, advanced data organization, help in data analysis, and displaying information and results in different ways. In this software, textual data are coded and classified through systematic processes. The use of software for coding qualitative data provides the possibility to simultaneously code codes, sub-codes and parts. Interviews and documents should be available in one space and be easily moved.

These items were added and highlighted in the method.

Comment:

• Ethical considerations need to be mentioned in detail.

Answer:

The study protocol was reviewed and approved by the Research Ethics Committees of the School of Public Health & Neuroscience Research Center - Shahid Beheshti University of Medical Sciences, Tehran. Iran (Approval ID: R. SBMU.PHNS.REC.1401.083).

The study was conducted in a quiet place.

Each worker was interviewed individually.

The confidentiality of workers' information and their opinions was maintained.

These items were added and highlighted in the method.

Comment:

• The abstract needs to be organized and completed.

Answer:

The abstract was organized and completed.

Introduction : accident of falling from a height is high among construction workers. Construction workers do not use harnesses. Thus, the present study was conducted to identify the factors affecting the non-use of harnesses among construction workers in Tehran, Iran.

Materials and methods In this study was conducted by interviewing professors and construction workers in order to identify factors affecting the non-use of harness. Factors influencing the non-use of safety harnesses were identified from the workers' point of view. The obtained data were classified and coded using MAXQDA 10 software. After that, the most essential, effective and powerful factors were identified using the degree and intersectionality of social network analysis.

Results: According to the interview results, 27 factors were determined as factors affecting non-use of harnesses by construction workers and divided into four main groups. The four groups were harness design, management factors, harness comfort, and attitudinal factors. Based on the results of the degree centrality, the non-ergonomic design and attitude of the harness inefficiency were identified as the most influential and powerful factors. The betweenness indicator also showed that the non-ergonomic design could mediate other factors in the non-use of the harness.

Conclusion: The findings showed that by considering various factors such as considering more comfort in the design of the ergonomic harness, it produced a better product. Also, the use of safety harnesses by workers increases.

These items were added and highlighted in the abstract.

Comment:

• The text needs to be edited by a language expert for ease of reading.

Answer:

The manuscript was re-edited by a language expert.

Thank you for your valuable comments.

Sincerely Yours

Dr. Mousa Jabbari

---

## [Editor Report · Decision Letter 2]

15 Jun 2023

Identification and classification of factors affecting the non-use of safety harness at height among construction workers in Tehran

PONE-D-23-00199R2

Dear Dr. Jabbari,

We’re pleased to inform you that your manuscript has been judged scientifically suitable for publication and will be formally accepted for publication once it meets all outstanding technical requirements.

Kind regards,

Mohammad Hossein Ebrahimi

Academic Editor

PLOS ONE
---

## [Editor Report · Acceptance letter]

18 Jul 2023

PONE-D-23-00199R2 

Identification and classification of factors affecting the non-use of safety harness at height among construction workers in Tehran 

Dear Dr. Jabbari:

I'm pleased to inform you that your manuscript has been deemed suitable for publication in PLOS ONE. Congratulations! Your manuscript is now with our production department. 

Kind regards, 

on behalf of

Dr. Mohammad Hossein Ebrahimi 

Academic Editor

PLOS ONE